# Isorhamnetin and Hispidulin from *Tamarix ramosissima* Inhibit 2-Amino-1-Methyl-6-Phenylimidazo[4,5-*b*]Pyridine (PhIP) Formation by Trapping Phenylacetaldehyde as a Key Mechanism

**DOI:** 10.3390/foods9040420

**Published:** 2020-04-03

**Authors:** Xiaopu Ren, Wei Wang, Yingjie Bao, Yuxia Zhu, Yawei Zhang, Yaping Lu, Zengqi Peng, Guanghong Zhou

**Affiliations:** 1Key Laboratory of Meat Processing and Quality Control, Ministry of Education China, Jiangsu Collaborative Innovation Center of Meat Production and Processing, Quality and Safety Control, College of Food Science and Technology, Nanjing Agricultural University, Nanjing 210095, China; alarxp@126.com (X.R.); 2018108046@njau.edu.cn (W.W.); 2015208017@njau.edu.cn (Y.B.); 2016208019@njau.edu.cn (Y.Z.); zhangyawei@njau.edu.cn (Y.Z.); ghzhou@njau.edu.cn (G.Z.); 2Xinjiang Production & Construction Group Key Laboratory of Agricultural Products Processing in Xinjiang South, College of Life Science, Tarim University, Alar 843300, China; 3College of Life Science, Nanjing Agricultural University, Nanjing 210095, China; lyphwq@njau.edu.cn

**Keywords:** 2-amino-1-methyl-6-phenylimidazo[4,5-*b*]pyridine (PhIP), flavonoid, flavonoid-phenylacetaldehyde adducts, model system, roast lamb patties

## Abstract

*Tamarix* has been widely used as barbecue skewers to obtain a good taste and a unique flavor of roast lamb in China. Many flavonoids have been identified from *Tamarix*, which is an important strategy employed to reduce the formation of heterocyclic amines (HAs) in roast meat. Isorhamnetin, hispidulin, and cirsimaritin from *Tamarix ramosissima* bark extract (TRE) effectively inhibit the formation of 2-amino-1-methyl-6-phenylimidazo[4,5-*b*] pyridine (PhIP), the most abundant HAs in foods, both in roast lamb patties and in chemical models. According to the results of the GC-MS analysis, TRE and the three flavonoids significantly reduced the contents of phenylacetaldehyde, an important intermediate involved in PhIP formation at three levels. A subsequent ultra performance liquid chromatography-mass spectrometry (UPLC-MS) analysis revealed that these flavonoids trapped phenylacetaldehyde by forming interaction adducts. The formation of three postulated adducts, 8-*C*-(*E*-phenylethenyl)isorhamnetin, 6-*C*-(*E*-phenylethenyl)isorhamnetin and 8-*C*-(*E*-phenylethenyl)hispidulin, in the chemical models and roast lamb patties was further confirmed by a TOF-MS/MS analysis. Our results demonstrate that TRE and the corresponding flavonoids trap phenylacetaldehyde to form adducts and thus inhibit PhIP formation, suggesting their great potential beneficial effects on human health.

## 1. Introduction

*Tamarix* is considered as an interesting potential source of functional food due to its abundant natural flavonoids [1]. Many flavonoids have been identified from various species of *Tamarix*, such as quercetin, isoquercetin, apigenin, amentoflavone, (+)-catechin, rhamnocitrin, isorhamnetin, tamarixetin, chrysoeriol and rhamnazin [1,2,3,4]. Hispidulin and cirsimaritin were firstly identified among 13 flavonoids detected in the genus *Tamarix* from southern Xinjiang, along with isorhamnetin, which are the main three flavonoids present in *T. ramosissima* (Figure 1) [5]. *Tamarix* has been widely used as barbecue skewers to obtain a good taste and a unique flavor of roast lamb in China, specifically in southern Xinjiang. However, the roasting of meat generates heterocyclic amines (HAs), and the International Agency for Research on Cancer (IARC) classified processed meat as “carcinogenic to humans” (group 1), based on more than 800 epidemiological studies that reported a link between meat consumption and cancer. Concerning roasted and barbecued meat, HAs were pointed out as components with high carcinogenic potential [6], of which 2-amino-1-methyl-6-phenylimidazo[4,5-*b*]pyridine (PhIP) is well known as the most abundant HA in roast meat, with levels of up to 480 ng/g detected [7]. PhIP is also the largest contributor to the estimated average daily HA exposure in China (29.06 ng/day, 58%) [8]. PhIP is regarded as a possible human carcinogen (class 2B) by IARC [9] and is also listed as reasonably anticipated to be a human carcinogen by the National Toxicology Program [10]. The factors that affected the formation of PhIP in foods including the cooking methods, cooking temperature, cooking time, the amounts of precursors and the content of substances with enhancing or inhibiting effects on the formation of PhIP [11].

Many strategies have been employed to reduce the formation of PhIP. Many works have demonstrated that HAs can be inhibited by natural antioxidants, such as spices and extracts rich in polyphenols [12,13,14]. However, the mechanisms underlying the effect have been overlooked. In the present work, the mechanism of action was explored more in-depth. Several flavonoids exhibited effective inhibition on PhIP. According to Zhu et al. [15], apigenin, luteolin, kaempferol, naringenin, quercetin, genistein, phlorizin and epigallocatechin gallate (EGCG) added to roast beef patties significantly inhibited PhIP formation and reduced its concentration by 60–80%. Norartocarpetin, quercetin and naringenin effectively inhibit PhIP formation in roast beef patties [16]. Naringenin and quercetin directly react with phenylacetaldehyde, an important intermediate involved in PhIP formation, to form interaction adduct, thus leading to a significant reduction in PhIP levels in model systems and real meat samples. Phenylacetaldehyde scavenging is considered a key mechanism of naringenin and quercetin [15,17]. Isorhamnetin is a 3′-*O*-methylated metabolite of quercetin. Hispidulin and cirsimaritin exhibit similar structural characteristics to naringenin. Based on the inhibitory effects of flavonoids on PhIP formation that depend upon the structural characteristics, researchers have hypothesized that isorhamnetin, hispidulin and cirsimaritin from *T. ramosissima* effectively inhibit PhIP formation, which has rarely been reported in real meat systems. The present study aimed to investigate the abilities of *T. ramosissima* bark extract (TRE), isorhamnetin, hispidulin and cirsimaritin to inhibit the formation of PhIP in the model system and roast lamb patties and to explore the probable inhibitory mechanism.

## 2. Materials and Methods 

### 2.1. Reagents and Materials

Phenylalanine, glucose, creatinine, diethylene glycol and phenylacetaldehyde were purchased from Yuanye Bio-Technology Co., Ltd. (Shanghai, China). Isorhamnetin, hispidulin and cirsimaritin were obtained from Sigma-Aldrich (Shanghai, China). The PhIP standard was purchased from Toronto Research Chemicals (Downsview, Ontario, Canada). Oasis MCX (60 mg, 3 mL) cartridges were supplied by Waters (Milford, MA, USA) and diatomaceous earth was purchased from Merck Co. (Darmstadt, Germany). All other chemicals and solvents used in this study were of HPLC or analytical grade. TRE was prepared using our previously reported procedure [5].

### 2.2. Determination of Additive Levels in Lamb Patties and Model Systems

The levels of TRE added to lamb patties were determined using the Standards for the Use of Food Additives (GB 2760-2014), with a maximum level of 0.50 mg/g of antioxidants in roast meat products; three levels (Level_p_ 1–3) of TRE were established in the present study (Table 1). According to Ren et al. [5], the concentrations of isorhamnetin, hispidulin and cirsimaritin were 36.91, 28.79 and 13.35 mg/g TRE, respectively, and the approximate levels of each compound are presented in Table 1 (Level_p_ 1–3). The concentrations of phenylalanine and creatinine in raw lamb were determined (1307 and 870 mg/kg, respectively) in the present study but were approximately 25 times lower than the concentrations in the PhIP model. For consistency, the concentrations of TRE and the three flavonoids were also increased 25 times, and the levels are listed in Table 1 (Level_m_ 1–3).

### 2.3. Preparation of Roast Lamb Patties

Lamb tenderloin was obtained from a local market in Nanjing, China. The levels of TRE, isorhamnetin, hispidulin and cirsimaritin were added as shown in Table 1, and ground lamb without additives was used as a control. After thorough mixing, each ground lamb mixture and the control lamb were incubated for 4 h at 4 °C and formed into lamb patties (20.0 g) using a petrified dish (6.0 cm diameter × 1.5 cm depth) to ensure uniformity. The patties were grilled in an electric oven (D3-256A, Toshiba, Japan) at 200 °C for 20 min (10 min per side) with a core temperature of 72.1–73.8 °C. The cooking losses of different treatments were 52.18–53.00% without significant difference among them. After cooking, the patties were cooled at room temperature and cut into small pieces. The pieces were freeze-dried and the experiment was performed in triplicate. Using the method reported by Zeng et al. [18], the roast lamb samples (5.0 g) were suspended in 30 mL of 1 mol/L NaOH and homogenized by magnetic stirring for 2 min at room temperature. The viscous solution was mixed with diatomaceous earth (13 g), ethyl acetate (50 mL) was added and the mixture was ultrasonically extracted for 30 min twice. The concentrated eluent was transferred to Waters Oasis MCX cartridges. The cartridges were then sequentially rinsed with 6 mL of 0.1 M HCl, methanol and a methanol-ammonia mixture (19:1, *v/v*) and then concentrated under nitrogen flushing. The residue was dissolved in 250 μL of methanol and filtrated through a 0.22 μm syringe filter immediately before the UPLC-MS analysis.

### 2.4. Analysis of the Effects of TRE and Flavonoids on PhIP Formation in Chemical Models

The effects of different concentrations of TRE and the three flavonoid compounds on PhIP formation were examined using a previously described method [17,19]. Mixtures of 2 mmol phenylalanine, 2 mmol creatinine and 1 mmol glucose were dissolved in 10 mL of diethylene glycol containing 14% water. The levels of TRE, isorhamnetin, hispidulin and cirsimaritin are presented in Table 1. These additives were directly added to the reaction vials and mixed thoroughly. The mixtures were heated at 200 °C in closed test tubes for 1 h and the temperature only fluctuated within 2 °C. After the heating process, the tubes were immediately cooled at room temperature for 5 min and −20 °C for 10 min to terminate the reaction and then samples were prepared for the UPLC-MS analysis. The experiment was performed in triplicate.

### 2.5. UPLC-MS Analysis of PhIP

The PhIP levels in the lamb patties were analyzed using the methods reported by Zheng et al. [16], with a slight modification. PhIP was identified and quantified using an Acquity UPLC BEH C18 column (1.7 μm; 2.1 mm × 100 mm I.D.; Waters) at 35 °C. Gradient elution was achieved with a binary mobile phase of 10 mmol/L ammonium acetate (pH 6.8) (A) and acetonitrile (B). The solvent composition was 0–0.1 min, 90% A; 0.1–18 min, 10–30% B; 18–20 min, 30–100% B; and 20–20.1 min, 100–10% B; the total flow rate was 0.3 μL/min. The injection volume was 2 μL. The following mass spectrometry parameters were used: positive ion mode; capillary voltage, 3.5 kV; ion source temperature, 120 °C; and desolvation temperature, 400 °C. Data were acquired and processed using Masslynx 4.1 software (Waters, Milford, MA, USA). The recovery (%) of PhIP in meat samples and model systems were 73.40% and 81.75%, respectively, and the LOD, LOQ, RSD and R^2^ value were 0.08 ng/g, 0.112 ng/g, 9.86% and 0.9982, respectively.

### 2.6. Evaluation of the Phenylacetaldehyde-Scavenging Capabilities of TRE and the Flavonoids in a Model System Using GC-MS

For the analysis of phenylacetaldehyde levels in the PhIP model system, we used the method described in a previous study [15]. Ten milliliters of reaction mixtures with different concentrations of TRE and three flavonoids were extracted twice with 20 mL of ethyl acetate. After thorough vortexing for 2 min, the ethyl acetate extract was dried with anhydrous sodium sulfate, filtered and then subjected to the GC-MS analysis. The experiment was performed in triplicate. The samples were analyzed using an Agilent gas chromatograph (Agilent 7890) coupled with an Agilent mass spectrometer (5975 MSD) with a triple-axis. Separation was performed using an HP-5 capillary column. Analyses were performed using the following parameters: inlet temperature, 200 °C; column flow, 1 mL/min (He); temperature program, 40 °C for 4 min, an increase of 5 °C/min up to 230 °C and a hold for 5 min; and injection volume, 1.0 μL. The correlation coefficient (*R*^2^) of the phenylacetaldehyde standard curve was 0.9895.

### 2.7. Detection of Flavonoid-Phenylacetaldehyde Adducts in Roast Lamb Patties

The extraction and detection of flavonoid-phenylacetaldehyde adducts in roast lamb was performed using the method described by Zhu et al. [15], with a few modifications. Roast lamb samples (20.0 g) were mixed with MilliQ water (50 mL) and homogenized for 5 min. Afterwards, the cloudy suspensions were extracted with ethyl acetate (200 mL) twice. The concentrated eluate was then evaporated to dryness and defatted with 2 mL of methanol. Finally, the mixtures were centrifuged and filtered before analysis using UPLC-MS.

### 2.8. Detection of Flavonoid-Phenylacetaldehyde Adducts in the Model System

The direct reactions between phenylacetaldehyde and the three flavonoids present in TRE in the model system was analyzed using the method reported by Cheng et al. [17], with some minor modifications. Phenylacetaldehyde was mixed with isorhamnetin, hispidulin and cirsimaritin in a molar ratio of 1:5 in 10 mL of diethylene glycol containing 14% water. The mixtures were vortexed thoroughly and heated at 200 °C for 1 h to ensure consistency with the conditions of the PhIP model system. The reaction mixtures were subsequently cooled at room temperature for 5 min and −20 °C for 20 min to terminate the reaction. The extraction solvent system consisted of a 1:1:2:1 (*v/v*) ratio of samples:MilliQ water:ethyl acetate:hexane. The mixture was vortex-extracted for 2 min and centrifuged at 8000 g for 10 min. Finally, the supernatant was collected and concentrated with a rotary evaporator under a vacuum. The extract was redissolved in methanol before analysis using the Waters UPLC-Q-TOF system. The experiment was performed in triplicate.

### 2.9. UPLC-MS Analysis of Flavonoid-Phenylacetaldehyde Adducts

The flavonoid-phenylacetaldehyde adducts formed in roast lamb patties and model systems were detected using an LC-MS system (G2-XS Q-TOF, Waters) equipped with a Waters Acquity UPLC. Samples were separated on an Acquity UPLC BEH C18 column (100 mm × 2.1 mm, 1.7 μm). The mobile phase consisted of 0.1% formic acid in water (buffer A), and methanol (buffer B). The elution gradient was 5% buffer B for 0.5 min, 5–95% buffer B for 11 min, and 95% buffer B for 2 min at a flow rate of 0.4 mL/min. Mass spectrometry was performed using an electrospray source in positive ion mode with MSe acquisition mode, with a selected mass range of 50–1200 *m/z*. The following ionization parameters were used: a capillary voltage of 2.5 kV, collision energy of 40 eV, source temperature of 120 °C, and desolvation gas temperature of 400 °C. Data were acquired and processed using Masslynx 4.1 software (Waters, Milford, MA, USA).

### 2.10. Statistical Analysis

Statistical analyses were conducted using the IBM SPSS Statistics program ver. 22 (SPSS Inc., Chicago, IL, USA). An ANOVA and Duncan’s test were used to assess the differences between different treatments and a linear regression was used to analyze the correlations between the phenylacetaldehyde-scavenging capabilities and the inhibitory effects on PhIP formation. Experiments were conducted in triplicate and data are reported as means ± SD. *p* < 0.05 was selected as the level for significant differences.

## 3. Results and Discussion

### 3.1. Effects of Additives on PhIP Formation in Lamb Patties and Model Systems

A UPLC-MS analysis was used to detect the concentrations of PhIP in lamb patties and model systems treated with and without the additives. The content of PhIP ranged from 0.12 to 0.52 ng/g in the patties, consistent with the values reported by Sun et al. [20]. The effects of TRE and three flavonoids on inhibiting PhIP formation in lamb patties and model systems are shown in Figure 2 and Figure 3. The ANOVA indicated significant effects of additive levels and the interaction between additives and additive levels on PhIP formation (*p* < 0.001). The addition of all of these four additives at three levels significantly inhibited the formation of PhIP in roast lamb patties and model systems (*p* < 0.05), and dose-dependent effects were observed for the TRE, isorhamnetin and hispidulin treatments.

In the lamb patties, the concentrations of PhIP were 0.36, 0.24 and 0.14 ng/g as the TRE levels increased from 0.15 to 0.45 mg/g (Figure 2), respectively. Significant differences in PhIP contents were observed between any two of the levels (*p* < 0.05), indicating that TRE significantly reduced PhIP formation. Similar inhibitory effects were also observed for isorhamnetin, and hispidulin, as the PhIP contents ranged from 0.16 to 0.34 ng/g of meat. However, no significant differences in the inhibition of PhIP formation were observed among different levels of cirsimaritin (*p* > 0.05). A treatment with 0.45 mg/g TRE produced the strongest inhibitory effects on PhIP formation among all treatments (*p* < 0.05), reducing the level of PhIP formed by 72.92%, followed by 18.0 μg/g isorhamnetin with an inhibition rate of 69.82%. However, no significant differences were observed between the two groups (*p* > 0.05). Compared to the TRE treatments, cirsimaritin exerted a significantly stronger inhibitory effect on PhIP formation at Level_p_ 1 (*p* < 0.05), whereas at Level_p_ 3, a significantly higher inhibition rate was observed for TRE (*p* < 0.05).

Similar inhibitory effects were observed on PhIP production in the model system. As shown in Figure 3, the inhibition rate of 12.5 mg/mL TRE was 62.15%, a value that was significantly lower than 25.0 mg/mL (69.11%) and 37.5 mg/mL (72.85%) TRE (*p* < 0.05). Treatment with the three flavonoids at different levels also exerted significant inhibitory effects on PhIP production (*p* < 0.05). When higher concentrations of isorhamnetin and hispidulin were added to the models, less PhIP was generated, whereas no significant difference was observed between treatments with different cirsimaritin levels (*p* > 0.05). Among all the treatments, 25.0 mg/mL and 37.5 mg/mL TRE resulted in larger inhibition rates of up to 69.11% and 72.85%, respectively, values that were significantly higher than all of the other treatments (*p* < 0.05). Notably, 0.03 mmol hispidulin produced the weakest inhibitory effect on PhIP formation (*p* < 0.05), with an inhibition rate of 47.74%. Isorhamnetin exerted a significantly greater suppressive effect on the PhIP formation at three levels than hispidulin (*p* < 0.05). As shown in the study by Zhang et al. [21], isorhamnetin is considered an effective inhibitor of PhIP formation in a model system, with an IC_50_ (50% inhibiting concentration) values of 1.675 mg/mL. In addition, Yu et al. [22] analyzed the creatinine/phenylalanine chemical model system and reported an IC_50_ value of isorhamnetin of 2.246 mg/mL. Similar to the present study, 0.03 mmol isorhamnetin (0.95 mg/mL) showed an inhibitory rate of 57.84% and the minor difference may be due to the heating conditions and the concentrations of precursors.

In the present study, TRE exhibited similar inhibitory effects on PhIP formation in lamb patties and the model system, consistent with the inhibitory effects of several plant extracts on PhIP formation reported in previous studies. An apple extract exerted a strong inhibitory effect on the formation of PhIP both in fried beef patties and in the chemical model system, and proanthocyanidins and phloridzin were identified as the dominant inhibitors of PhIP generation [23]. Zhang et al. [21] reported an effective inhibition of PhIP formation following the addition of antioxidant of bamboo leaves (AOB) to both fried pork and a chemical model system, and the main four flavonoids present in AOB, orientin, homoorientin, vitexin and isovitexin, all exerted significant dose-dependent inhibitory effects on PhIP formation. Isorhamnetin, hispidulin and cirsimaritin were the main inhibitors of PhIP present in TRE. Many other flavonoids have been reported to inhibit PhIP formation in chemical models and real food systems. According to Cheng et al. [24], 0.2 mmol of quercetin produces a 77% reduction in the PhIP level compared to the control; naringenin and EGCG also significantly reduce the level of PhIP. Fan et al. [25] further confirmed the efficient inhibitory effect of quercetin on PhIP formation and that the inhibition rates depended on the dose. Moon et al. [26] reported superior inhibitory effects of 1000 ppm EGCG and naringenin on PhIP formation in a model system, with 99.2% and 97.6% of PhIP inhibition rates, respectively.

The differences in the inhibitory effects of isorhamnetin, hispidulin and cirsimaritin on PhIP formation are likely based on their diverse structures and substituent groups. From the structure-activity perspective, Salazar et al. [27] described the relationships between the inhibitory effects on PhIP and structural characteristics of twenty-five phenolic compounds. In the present study, isorhamnetin, hispidulin and cirsimaritin significantly inhibited PhIP formation both in roast lamb patties and in chemical models mainly through the three mechanisms listed below. The first and foremost is the presence of two hydroxy groups at meta positions of the A ring in isorhamnetin and hispidulin. The introduction of a methoxy group at the C-6 position on the A ring of hispidulin slightly decreases the inhibitory effect compared to isorhamnetin. Cirsimaritin contains a 3-methoxyphenol group, which is also a very efficient PhIP inhibitor. Second, the presence of one hydroxy group at the C-4′ position of B ring in the three flavonoids contributed to their substantial inhibition of PhIP formation. Third, the presence of the conjugated carbonyl group on the C ring increases the inhibitory effect of the three flavonoids on PhIP formation. Yu et al. [22] also confirmed that the hydroxyl groups on the A ring of the detected flavonoids, particularly at the ortho positions (C-5 and C-7), inhibited PhIP formation to the greatest extent. The hydroxyl groups on the B and C rings of the flavonoids may exert a certain synergistic effect with the C-5 and C-7 hydroxyl groups on the A ring, which increased the inhibitory effect on PhIP formation.

Generally, TRE and the three flavonoids exhibited similar activity in inhibiting PhIP formation in lamb patties and the model system, rather than a superposition of the inhibitory effects of individual flavonoids, which may be due to the antagonistic effects of flavonoids in TRE. The antagonistic or synergistic effects of polyphenolic compounds on HA formation have also been observed by other researchers. As shown in the study by Zeng et al. [18], rutin and protocatechuic acid exert antagonistic effects on DMIP and 4,8-DiMeIQx, while synergistic effects are observed for harman, norharman and MeIQx. Two other phenolic compounds, *p*-coumaric acid and ferulic acid, exert antagonistic effects on 4,8-DiMeIQx but synergistic effects on MeIQx. Lee et al. [28] also indicated that the antagonistic or synergistic effects of various antioxidants reduce or increase HA formation, depending on the types and concentrations of antioxidants.

### 3.2. Phenylacetaldehyde-Scavenging Capabilities of TRE and Flavonoids in the Model System

Phenylacetaldehyde, a Strecker or thermal degradation product of phenylalanine, plays an important role in the mechanism of PhIP formation, according to Zochling et al. [29]. Several previous studies have proposed that the inhibitory effects of numerous flavonoids on PhIP formation are attributed to trapping phenylacetaldehyde via the formation of flavonoid-phenylacetaldehyde adducts [15,16,17,30]. The ability of a flavonoid to directly scavenge or trap phenylacetaldehyde is considered the key step to reduce the yield of PhIP. In our present study, the ANOVA indicated significant effects of additives, additive levels and the interaction between additives and additive levels on PhIP formation (*p* < 0.05), as shown in Figure 4. GC-MS quantification of phenylacetaldehyde levels did not reveal significant differences in the phenylacetaldehyde-trapping abilities of hispidulin and cirsimaritin when the compounds were added at three levels (*p* > 0.05). The residual phenylacetaldehyde contents of the hispidulin and cirsimaritin groups ranged from 0.07 to 0.09 mmol/L, with trapping ratios ranging from 43.66% to 60.03%. Additionally, a significant difference was not observed between isorhamnetin and hispidulin or cirsimaritin at Level_m_ 3 (*p* > 0.05). The consistent phenylacetaldehyde-trapping capabilities of the three flavonoids are potentially attributed to their similar structural characteristics. Several essential structures of flavonoids are pivotal to increase the reactive carbonyl species (RCS)-trapping efficacy, including the hydroxyl group at C-5 on the A ring and the double bond between C-2 and C-3 on the C ring, but the trapping efficacy was not significantly altered by the number of hydroxyl groups on the B ring [31]. All three flavonoids, isorhamnetin, hispidulin and cirsimaritin, contain a hydroxyl group at C-5 on the A ring and the double bond between C-2 and C-3 on the C ring, resulting in the excellent phenylacetaldehyde-scavenging abilities. According to Cheng et al. [17], naringenin effectively reduces the phenylacetaldehyde level in a dose-dependent manner, with a trapping ratio of approximately 60%. Zhu et al. [15] suggested that all detected flavonoids, including apigenin, luteolin, genistein, EGCG, quercetin, naringenin and kaempferol, exhibited strong abilities to trap phenylacetaldehyde, with trapping ratios ranging from 21.85% to 88.81%. These findings correspond well with the relationship between the structures of flavonoids and their phenylacetaldehyde-scavenging abilities.

As shown in Figure 4, TRE exhibited the strongest ability to trap phenylacetaldehyde of the compounds analyzed in the present study, and the residual concentrations of phenylacetaldehyde in model systems following treatment with three different levels were 0.02, 0.03 and 0.02 mmol/L, with trapping ratios of 87.54%, 82.16% and 86.57%, respectively. Significant difference was not observed among the three levels of TRE (*p* > 0.05). The phenylacetaldehyde-trapping ratios of the three TRE levels were approximately two times higher than isorhamnetin, hispidulin and cirsimaritin (*p* < 0.05), and the stronger abilities of TRE may be attributed to the synergistic effects of the different flavonoids in TRE, as described by Shao et al. [31], who clearly showed that different flavonoids trap reactive dicarbonyl species with a synergistic effect when added at different concentrations. Furthermore, these researchers proposed that the reduction in the phenylacetaldehyde concentration induced by different flavonoids might be attributed to the reactions between phenylacetaldehyde with flavonoids to form different adducts. Some flavonoid-phenylacetaldehyde adducts may form between phenylacetaldehyde and isorhamnetin and hispidulin and cirsimaritin in the chemical system and roast lamb patties.

### 3.3. Correlations between the Phenylacetaldehyde-Trapping and PhIP-Inhibiting Abilities of TRE and Flavonoids

The correlations between the phenylacetaldehyde-scavenging capabilities of TRE and three flavonoids and their inhibitory effects on PhIP formation in both the chemical models and roast lamb patties were analyzed using a linear regression model. Significant correlations were observed between the PhIP-inhibiting and phenylacetaldehyde-scavenging capabilities of TRE, isorhamnetin and hispidulin. The strongest correlations were identified for the phenylacetaldehyde-scavenging and PhIP-inhibiting capabilities of TRE and isorhamnetin in chemical models and roast lamb patties, with *R*^2^ = 0.9050 and 0.9147 for TRE and *R*^2^ = 0.8495 and 0.9040 for isorhamnetin, respectively (*p* < 0.01, Figure 5A,B). Moreover, significant correlations were also observed for hispidulin with *R*^2^ = 0.7729 and 0.7710, respectively (*p* < 0.05, Figure 5C). Based on these results, the abilities of TRE, isorhamnetin and hispidulin to inhibit PhIP formation were strongly correlated with their abilities to trap phenylacetaldehyde. Thus, the main mechanism by which TRE, isorhamnetin and hispidulin inhibit PhIP formation in chemical models and roast lamb are through the trapping or reaction with phenylacetaldehyde, consistent with several previous studies [15,16,17,30,32,33]. However, poor correlations were observed for cirsimaritin (*p >* 0.05, Figure 5D). Some other mechanisms are likely responsible for the excellent inhibitory effects of cirsimaritin on PhIP formation, which require further investigation.

### 3.4. Analysis of Flavonoid-Phenylacetaldehyde Adducts in Roast Lamb Patties and Chemical Reaction Systems

Several previous studies have confirmed that many flavonoids inhibit PhIP formation by scavenging phenylacetaldehyde and forming the corresponding adducts [15,16,17,30,32,33]. In the present study, three flavonoid-phenylacetaldehyde adducts, 8-*C*-(*E*-phenylethenyl)isorhamnetin, 6-*C*-(*E*-phenylethenyl)isorhamnetin and 8-*C*-(*E*-phenylethenyl)hispidulin, were isolated from the chemical reaction system and identified using a UPLC-MS analysis (Figure 6). Flavonoid-phenylacetaldehyde adducts were identified in the model system and roast meat by separating the complexes with UPLC and comparing the retention times and molecular masses in the TOF-MS/MS data. The chromatograms of the reaction products of phenylacetaldehyde and flavonoids in the model system are shown in Figure 6. For isorhamnetin-phenylacetaldehyde adducts, two extracted molecular ion peaks at *m/z* 419.1 [M + H]^+^ were detected in positive ESI-MS and belonged to the molecular ion peaks of 8-*C*-(*E*-phenylethenyl)isorhamnetin) and 6-*C*-(*E*-phenylethenyl) isorhamnetin, whose molecular weight was 418, corresponding to the formation of adducts between isorhamnetin and phenylacetaldehyde, followed by the elimination of an H_2_O molecule. Regarding the hispidulin-phenylacetaldehyde adducts, due to the methoxy substitution at the C-6 position of the hispidulin A ring, only one molecular ion peak at *m/z* 403.1 [M + H]^+^ was detected in positive ESI-MS, which belonged to the adduct of 8-*C*-(*E*-phenylethenyl)hispidulin). Its molecular weight was 402, corresponding to the formation of an adduct between hispidulin and phenylacetaldehyde, followed by the elimination of an H_2_O molecule, suggesting that hispidulin reacted with phenylacetaldehyde and formed an adduct. However, in contrast to the former two flavonoids, no characteristic ion peak was observed for cirsimaritin at *m/z* 417.1 [M + H]^+^ in positive EIS-MS with a molecular weight of 416, implying that a direct adduction reaction between cirsimaritin and phenylacetaldehyde did not occur (Appendix A). The identified adducts in the model system were also observed in lamb patties that had been roasted for 20 min at 200 °C. These flavonoid-phenylacetaldehyde adducts were identified based on the molecular masses and TOF-MS/MS peaks. The chromatograms of adducts in roast lamb are shown in Figure 6B. The identification was also unobjectionably confirmed by comparing the data with related references [15,16,32].

The results of the collision-induced dissociation (CID) analysis showed the fragmentation patterns with diagnostic losses of 15, 32, 88, 164 and 268 for isorhamnetin adducts, and 15, 32, 92, 133 and 282 for hispidulin adduct (Figure 6C). A proposed fragmentation pathway resulting in the generation of the characteristic fragment ions is shown in Figure 7. The isorhamnetin and hispidulin adducts displayed fragment ions of *m/z* 404 and 388, corresponding to the loss of a methyl group from the B ring and A ring, respectively. The presence of fragment ions at *m/z* 255 and 270 was ascribed to the retro-Diels-Alder cleavage of the C ring from the ions observed at *m/z* 404 and 388. These fragmentation patterns and data from previous studies indicated that the adducts at *m/z* 419 and 403 mostly resulted from the electrophilic substitution of phenylacetaldehyde on the A ring of isorhamnetin at the C-6 or C-8 positions and hispidulin at the C-8 position [15,16,17,30,33]. Thus, the mechanism underlying the inhibitory effects of isorhamnetin and hispidulin on PhIP formation was mainly attributed to the trapping of phenylacetaldehyde, consistent with previous studies [15,16,17,30,33], but the mechanism underlying the inhibitory effects of cirsimaritin is not completely understood.

According to Navarro et al. [34] and Shao et al. [31], the A ring of the flavonoids is the major active site for trapping RCS, and the conjugation mainly occurs at positions C-6 and C-8 of the A ring. Lo et al. [35] found that EGCG was able to trap RCS, such as methylglyoxal (MGO), and generated new adducts at the C-8 position of the EGCG A ring; the trapping reactions also occurred between RCS and some other polyphenols at the C-6 or C-8 position of the A ring. Regarding the phenylacetaldehyde-trapping activities of flavonoids in the PhIP-producing system, more than 10 flavonoid-phenylacetaldehyde adducts have been isolated and identified with different flavonoids. Naringenin was first observed to substitute at either the C-6 or C-8 position of the A ring to produce 8-*C*-(*E*-phenylethenyl)naringenin and 6-*C*-(*E*-phenylethenyl)naringenin [17]. EGCG with 8 hydroxyl substituents, which are essential for the free radical scavenging functions of phenolic compounds, forms phenylethenyl-EGCG on the A ring (C-6 or C-8 position) [30]. Moreover, norartocarpetin, quercetin, kaempferol, apigenin, luteolin, pyridoxamine, dihydromyricetin and myricetin all exhibit phenylacetaldehyde-trapping activities, and C-6 and C-8 of the A ring of these flavonoids were identified as the active sites [15,16,25,32,33].

However, cirsimaritin exhibited potential abilities to inhibit PhIP formation and trap phenylacetaldehyde (Figure 2, Figure 3, and Figure 4), but the generation of the adduct at the C-8 position of the A ring was not detected in the present models (Appendix A). Navarro et al. [34] stated that in the model reaction, hydroxytyrosol (HT) was previously degraded into 3,4-dihydroxyphenylacetic acid (DOPAC), which directly reacted with MGO through an electrophilic aromatic substitution and generated DOPAC-MGO adducts and was responsible for the MGO trapping capacity. Zhou et al. [33] also reported that almost all the dihydromyricetin was degraded or transformed after a heat treatment at 128°C for 2 h, and the degraded or transformed product, namely, myricetin, trapped phenylacetaldehyde to form the adducts at positions C-6 or C-8 on the A ring of myricetin, thus inhibiting the generation of PhIP. Accordingly, we have hypothesized that cirsimaritin is degraded or transformed into some other compounds after heat treatments in a model system and roast lamb, and these new compounds may possess excellent phenylacetaldehyde-trapping and PhIP-inhibiting capacities, but the potential mechanism of action requires further study.

Some of these flavonoid-phenylacetaldehyde adducts exert potent beneficial effects on human health. As shown in the study by Li et al. [36], 6-*C*-(*E*-phenylethenyl)-naringenin binds to cyclooxygenase-1, which plays a critical role in human colorectal carcinogenesis, and specifically inhibits its activity both in vitro and ex vivo to finally suppress the growth of colorectal cancer cells. Zhao et al. [37] confirmed that 6-*C*-(*E*-phenylethenyl)-naringenin suppresses colon cancer cell proliferation and induces cytoprotective autophagy by inhibiting isoprenylcysteine carboxyl methyltransferase (Icmt)/RAS signaling pathways. Zhao et al. [38] revealed that 8-*C*-(*E*-phenylethenyl)quercetin inhibits colon cancer cell growth by inducing autophagic cell death through the activation of extracellular signal-regulated kinase (ERK). According to Zheng et al. [16], six flavonoid-phenylacetaldehyde adducts composed of naringenin, norartocarpetin and quercetin showed moderate cytotoxic activity toward different liver cancer cell lines, with IC_50_ values ranging from 10 to 40 μM, indicating that these compounds were much more potent than 5-fluorouracil (the positive control, IC_50_ > 100 μM). Among these adducts, 8-*C*-(*E*-phenylethenyl)norartocarpetin exerted the strongest inhibitory effect on liver cancer cells and displayed promising anticancer capabilities. Furthermore, the cytotoxicity of flavonoid-phenylacetaldehyde adducts on the cancer cells was considered to mainly depend on the flavonoids’ structures and the position at which the phenylacetaldehyde was attached [16]. Recently, the potent anticancer activities of isorhamnetin and hispidulin on colorectal cancer [39,40], skin cancer [41] and human clear cell renal cell carcinoma [42] have been reported. Hence, the flavonoid-phenylacetaldehyde adducts identified in the present study are also expected to display anticancer properties, which require further investigation.

## 4. Conclusions

In conclusion, TRE, isorhamnetin, hispidulin and cirsimaritin significantly inhibited the formation of PhIP in both roast lamb and chemical model systems (*p* < 0.05), with the largest inhibitory rate of 72.92%. These flavonoids effectively decreased the concentration of the key intermediate of PhIP formation, phenylacetaldehyde, by directly reacting with phenylacetaldehyde to form the corresponding adducts 8-*C*-(*E*-phenylethenyl)isorhamnetin, 6-*C*-(*E*-phenylethenyl)isorhamnetin and 8-*C*-(*E*-phenylethenyl)hispidulin, which were reported for the first time in the present study. Significant and strong correlations between the phenylacetaldehyde-trapping and PhIP-inhibiting capabilities of TRE, isorhamnetin and hispidulin were identified in roast lamb and chemical systems (*p* < 0.05). These findings make important contributions to our understanding of the mechanism underlying the inhibition of PhIP formation and provide useful information for reducing the generation of HAs in thermally processed meat.

## Figures and Tables

**Figure 1 foods-09-00420-f001:**
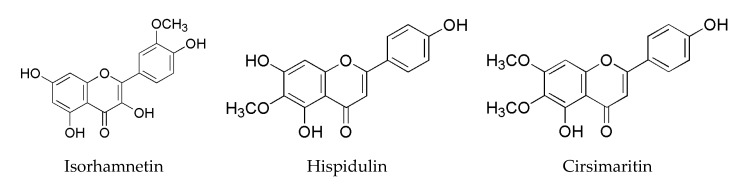
Structures of the three flavonoids derived from *Tamarix ramosissima* bark extract (TRE).

**Figure 2 foods-09-00420-f002:**
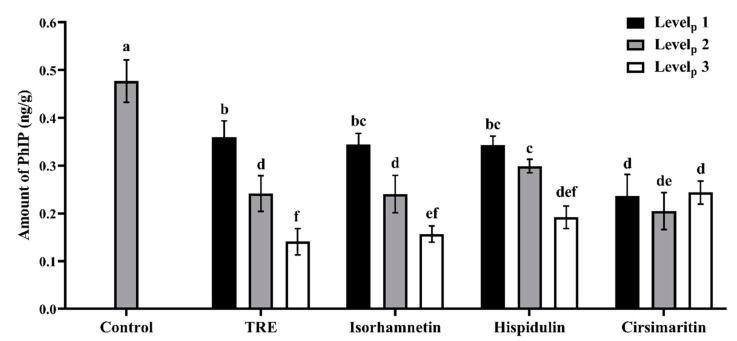
Inhibitory effects of different concentrations of TRE, isorhamnetin, hispidulin and cirsimaritin on the formation of PhIP in roast lamb patties (*n* = 3). Level_p_ represents the additive levels in patties. Different letters (a–f) above the error bars indicate significant differences (*p* < 0.05).

**Figure 3 foods-09-00420-f003:**
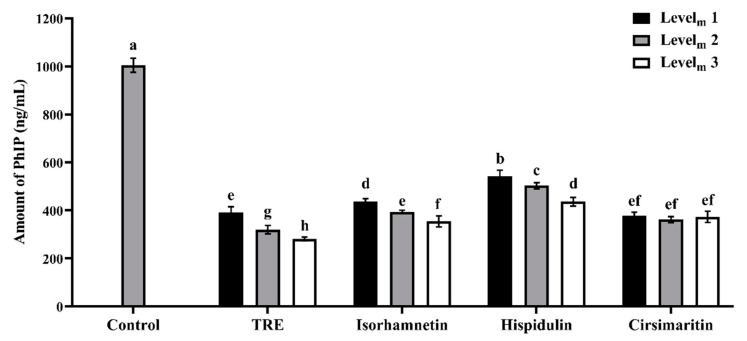
Inhibitory effects of different concentrations of TRE, isorhamnetin, hispidulin and cirsimaritin on the formation of PhIP in model systems (*n* = 3). Level_m_ represents additive levels in the model system. Different letters (a–h) above the error bars indicate significant differences (*p* < 0.05).

**Figure 4 foods-09-00420-f004:**
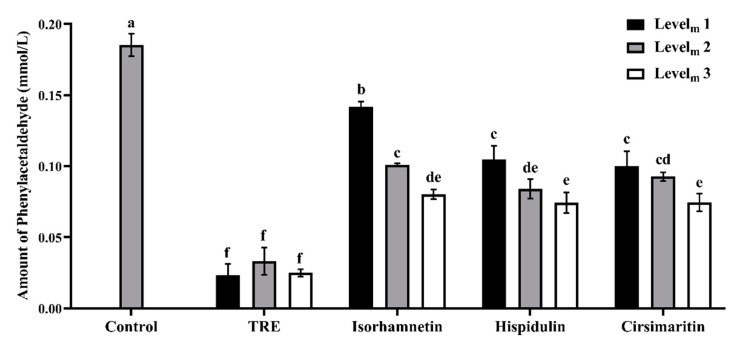
Effects of different concentrations of TRE, isorhamnetin, hispidulin and cirsimaritin on the amounts of phenylacetaldehyde detected in model systems (*n* = 3). Level_m_ represents additive levels in the model system. Different letters (a–f) above the error bars indicate significant differences (*p* < 0.05).

**Figure 5 foods-09-00420-f005:**
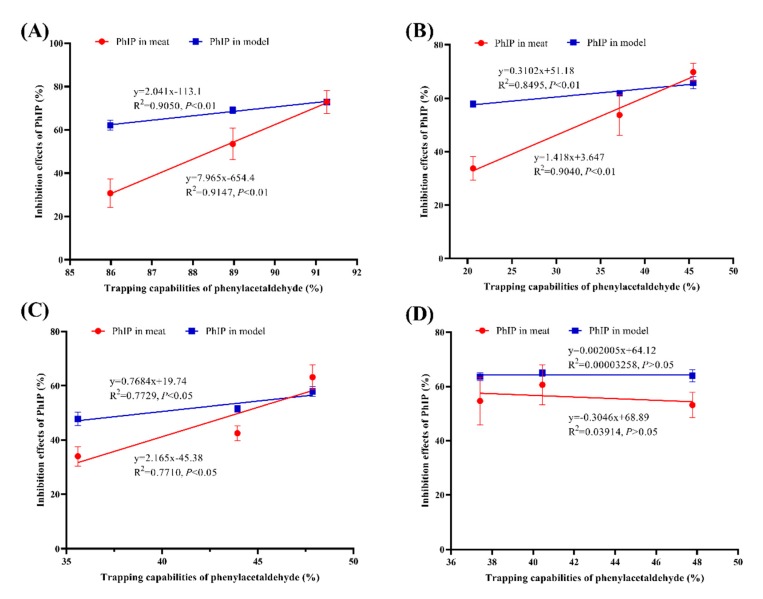
Correlations of different concentrations of TRE (**A**), isorhamnetin (**B**), hispidulin (**C**) and cirsimaritin (**D**) on the phenylacetaldehyde-scavenging rate and the inhibition rates of PhIP formation in model systems and roast lamb patties.

**Figure 6 foods-09-00420-f006:**
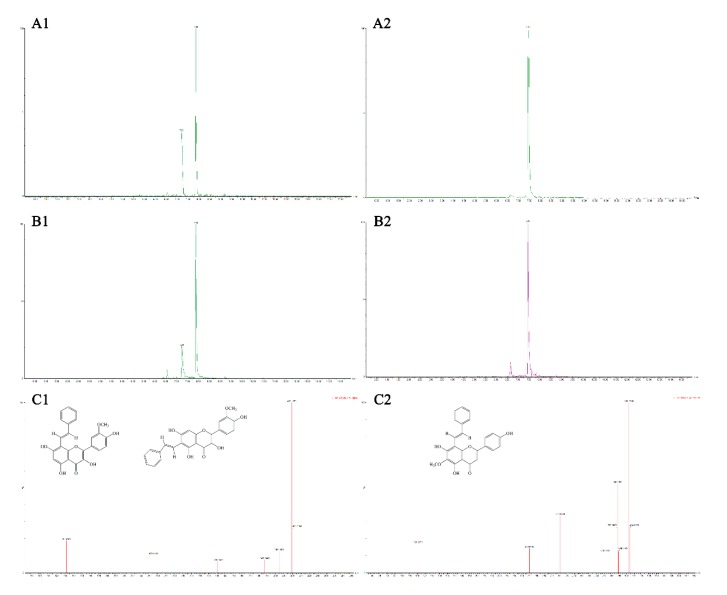
Flavonoid-phenylacetaldehyde adducts in chemical reaction systems and roast lamb patties. Extracted ion chromatograms of flavonoid-phenylacetaldehyde adducts in chemical reaction systems (**A**) and roast lamb patties (**B**); TOF-MS/MS spectrum of flavonoid-phenylacetaldehyde adducts (**C**).

**Figure 7 foods-09-00420-f007:**
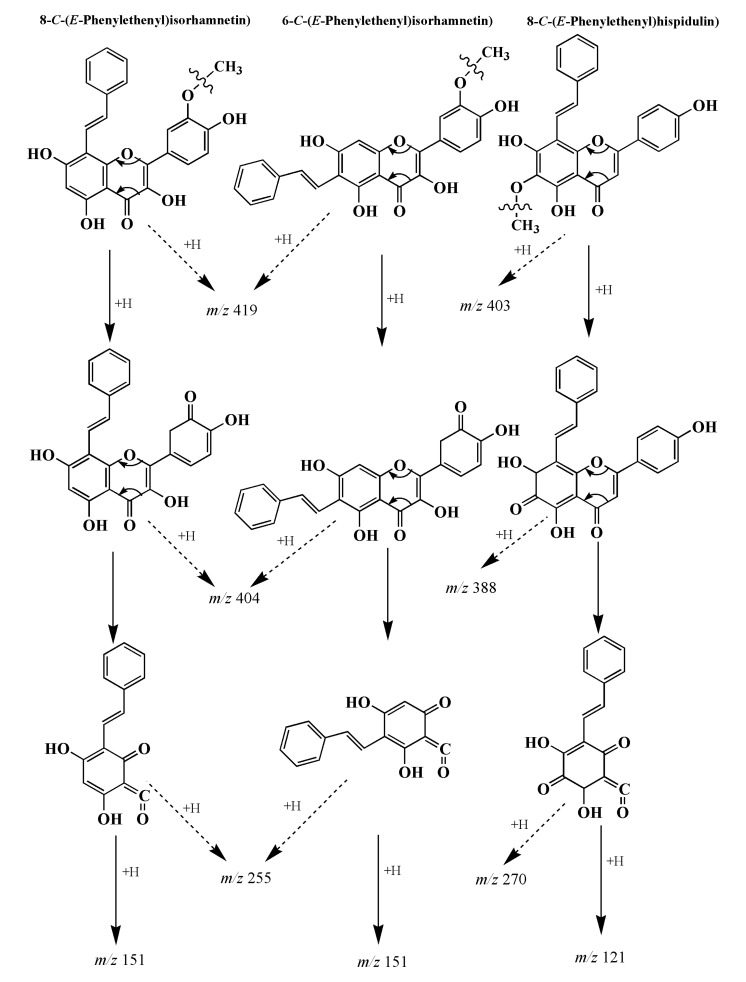
MS/MS fragmentation ions of the three adducts (*m/z* 419 and 403).

**Table 1 foods-09-00420-t001:** Levels of TRE and its three flavonoid compounds in lamb patties and the model system.

Groups	Lamb Patty			Model System		
	Level_p_ 1	Level_p_ 2	Level_p_ 3	Level_m_ 1	Level_m_ 2	Level_m_ 3
TRE	0.15 mg/g	0.30 mg/g	0.45 mg/g	12.5 mg/mL	25.0 mg/mL	37.5 mg/mL
Isorhamnetin	6.0 μg/g	12.0 μg/g	18.0 μg/g	0.03 mmol	0.16 mmol	0.32 mmol
Hispidulin	3.0 μg/g	6.0 μg/g	9.0 μg/g	0.03 mmol	0.08 mmol	0.16 mmol
Cirsimaritin	1.5 μg/g	3.0 μg/g	4.5 μg/g	0.02 mmol	0.03 mmol	0.06 mmol

Notes: Level_p_ represents the level of each additive in roast lamb patties. Level_m_ represents the additive level in the model system. The numbers (1, 2 and 3) represent the different additive levels.

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
