# Peer review of "Isorhamnetin and Hispidulin from Tamarix ramosissima Inhibit 2-Amino-1-Methyl-6-Phenylimidazo[4,5-b]Pyridine (PhIP) Formation by Trapping Phenylacetaldehyde as a Key Mechanism"

_foods, 2020, doi:10.3390/foods9040420_

Round 1
Reviewer 1 Report
Here, Ren et al investigated the abilities of T. ramosissima bark extract, and three flavonoids (isorhamnetin, hispidulin, and cirsimaritin) to inhibit the formation of PhIP in a chemical model system and roast lamb patties. Furthermore, they explored the probable inhibitory mechanism. Overall, the manuscript is clearly written and justifies why the investigations were warranted. Methods were generally clearly described and represented a fairly comprehensive investigation of the extract and related flavonoids. No major concerns were noted.
Author Response
Point 1: Here, Ren et al investigated the abilities of T. ramosissima bark extract, and three flavonoids (isorhamnetin, hispidulin, and cirsimaritin) to inhibit the formation of PhIP in a chemical model system and roast lamb patties. Furthermore, they explored the probable inhibitory mechanism. Overall, the manuscript is clearly written and justifies why the investigations were warranted. Methods were generally clearly described and represented a fairly comprehensive investigation of the extract and related flavonoids. No major concerns were noted.
Response 1: Thank you very much for your comments.
Reviewer 2 Report
The manuscript is interesting scientific contributions to the knowledge of the isorhamnetin and hispidulin from Tamarix ramosissima inhibit PhIP formation by trapping phenylacetaldehyde as a key mechanism. The objective of this study was to investigate the abilities of T. ramosissima bark extract, isorhamnetin, hispidulin, and cirsimaritin to inhibit the formation of PhIP in model system and roast lamb patties, and to explore the probable inhibitory mechanism. The paper has high scientific level, the experiment is well designed, the discussion is consistent and the final conclusions are interesting.
Suggestions for edition as well as some comments are the following:
Abstract
It´s Ok
Introduction
Please include the main factor that favor the heterocyclic amines formation
Material and methods
Line 97, I think that the condition used are not very usual (200 ºc for 20 minutes). I think that it´s long time.
Did you measure the temperature in the cuore of lamb patties?. This data is very important. Please include this information in the manuscript
Results and discussion
Lines 235, 236, 241, 251, etc.., please revise the refences according to the guide for authors.
Line 311 correlations. I recommend the authors to remove this part of the manuscript due to the low samples in this study
Conclusions
It´s Ok
References
It´s Ok
Figures and tables
It´s Ok
I hope that my comments can improve the manuscript.
Author Response
Point 1: Introduction, Please include the main factor that favor the heterocyclic amines formation.
Response 1: As Reviewer suggested that the main factors that favored the HAs formation were added in the manuscript at the end of first paragraph of Introduction section. Thank you very much for your comments.
Point 2: Material and methods, Line 97, I think that the condition used are not very usual (200 ºc for 20 minutes). I think that it´s long time. Did you measure the temperature in the cuore of lamb patties?. This data is very important. Please include this information in the manuscript
Response 2: The conditions used in this study (200℃ for 20 min) were referred to several studies. The roasting time (10 min per side) was quite common for roasting meat, especially for meat patties (Khan et al., 2019; Zeng et al., 2014, 2016, 2018; Chen et al., 2017; Zhu et al., 2016; Zheng et al., 2016). Regarding to the roasting temperature, 200℃ was optimal for PhIP formation because PhIP is unstable during heating and susceptible to degradation due to the more conjugated carbon-carbon double bonds and a benzene ring attached to a single bond of a side chain (Arvidsson et al., 1997; Chiu and Chen, 2000). Many researchers also selected 200℃ as the cooking and reaction temperature (Cheng et al., 2007; Wong et al.,2012; Bordas et al., 2004; Khan et al., 2019). The core temperature of the lamb patties was measured by a thermal probe and was 72.1-73.8℃ in the present roasting conditions. We have added this information in section of 2.3 of the manuscript (Line 102). Thanks very much for your comments.
References:
Khan, I. A.; Liu, D.; Yao, M.; Memon, A.; Huang, J.; Huang, M., Inhibitory effect of Chrysanthemum morifolium flower extract on the formation of heterocyclic amines in goat meat patties cooked by various cooking methods and temperatures. Meat Sci. 2019, 147, 70-81.
Zeng, M.; He, Z.; Zheng, Z.; Qin, F.; Tao, G.; Zhang, S.; Gao, Y.; Chen, J., Effect of six Chinese spices on heterocyclic amine profiles in roast beef patties by ultra performance liquid chromatography-tandem mass spectrometry and principal component analysis. J. Agr. Food Chem. 2014, 62, 9908-9915.
Zeng, M.; Li, Y.; He, Z.; Qin, F.; Chen, J., Effect of phenolic compounds from spices consumed in China on heterocyclic amine profiles in roast beef patties by UPLC-MS/MS and multivariate analysis. Meat Sci. 2016, 116, 50-57.
Zeng, M.; Wang, J.; Zhang, M.; Chen, J.; He, Z.; Qin, F.; Xu, Z.; Cao, D.; Chen, J., Inhibitory effects of Sichuan pepper (Zanthoxylum bungeanum) and sanshoamide extract on heterocyclic amine formation in grilled ground beef patties. Food Chem. 2018, 239, 111-118.
Chen, J.; He, Z.; Qin, F.; Chen, J.; Cao, D.; Guo, F.; Zeng, M., Inhibitory profiles of spices against free and protein-bound heterocyclic amines of roast beef patties as revealed by ultra-performance liquid chromatography-tandem mass spectrometry and principal component analysis. Food Funct. 2017, 11, 3938-3950.
Zhu, Q.; Zhang, S.; Wang, M.; Chen, J.; Zheng, Z., Inhibitory effects of selected dietary flavonoids on the formation of total heterocyclic amines and 2-amino-1-methyl-6-phenylimidazo[4,5-b]pyridine (PhIP) in roast beef patties and in chemical models. Food Funct. 2016, 7, 1057-1066.
Zheng, Z.; Yan, Y.; Xia, J.; Zhang, S.; Wang, M.; Chen, J.; Xu, Y., A phenylacetaldehyde-flavonoid adduct, 8-C-(E-phenylethenyl)-norartocarpetin, exhibits intrinsic apoptosis and MAPK pathways-related anticancer potential on HepG2, SMMC-7721 and QGY-7703. Food Chem. 2016, 197, 1085-1092.
Arvidsson, P.; Van Boekel, M. A.; Skog, K.; Jagerstad, M., Kinetics of formation of polar heterocyclic amines in a meat model system. J Food Sci. 1997, 5, 911-916.
Chiu, C. P.; Chen, B. H., Stability of heterocyclic amines during heating. Food Chem. 2000, 3, 267-272.
Cheng, K.; Wu, Q.; Zheng, Z.; Peng, X.; Simon, J.; Chen, F.; Wang, M., Inhibitory effect of fruit extracts on the formation of heterocyclic amines. J. Agr. Food Chem. 2007, 55, 10359-10365.
Wong, D.; Cheng, K.; Wang, M., Inhibition of heterocyclic amine formation by water-soluble vitamins in Maillard reaction model systems and beef patties. Food Chem. 2012, 133, 760-766.
Bordas, M.; Moyano, E.; Puignou, L.; Galceran, M.T., Formation and stability of heterocyclic amines in a meat flavour model system: effect of temperature, time and precursors. J Chromatogr B, 2004, 1, 11-17.
Point 3: Results and discussion, Lines 235, 236, 241, 251, etc.., please revise the refences according to the guide for authors. Line 311 correlations. I recommend the authors to remove this part of the manuscript due to the low samples in this study
Response 3: We are very sorry for our incorrect writing in the references mentioned above and we have corrected them according to the guide for authors in full text. In our opinions, it was necessary to demonstrate the correlations in the manuscript. They showed the relationships between the abilities of TRE, isorhamnetin and hispidulin to inhibit PhIP formation and their abilities to trap phenylacetaldehyde, which were closely related to the inhibitory mechanism on PhIP formation, although there were only three independent replicates due to the high test costs. The similar experiment design was also reported by several other researchers (Zhu et al., 2016; Cheng et al., 2008).
References
Zhu, Q.; Zhang, S.; Wang, M.; Chen, J.; Zheng, Z., Inhibitory effects of selected dietary flavonoids on the formation of total heterocyclic amines and 2-amino-1-methyl-6-phenylimidazo[4,5-b]pyridine (PhIP) in roast beef patties and in chemical models. Food Funct. 2016, 7, 1057-1066.
Cheng, K.; Wong, C.; Cho, C.; Chu, I.; Sze, K.; Lo, C.; Chen, F.; Wang, M., Trapping of phenylacetaldehyde as a key mechanism responsible for naringenin’s inhibitory activity in mutagenic 2-amino-1-methyl-6-phenylimidazo [4,5-b]pyridine formation. Chem. Res. Toxicol. 2008, 21, 2026-2034.
Reviewer 3 Report
The manuscript has interesting scientific contribution to the knowledge about inhibitory effect of Tamarix extract and three flavonoids on the formation most frequent and abundant heterocyclic aromatic amine (PhIP) both in grilled lamb patties and model systems. The topic is interest and the article is good and easy to follow. However some ideas and refrences should be introduced:
- In Introduction add the following idea: The International Agency for Research on Cancer (IARC) classified processed meat as “carcinogenic” to humans (group 1), based on > 800 epidemiological studies that reported a link between meat consumption and cancer. Concerning grilled and barbecued meat, heterocyclic aromatic amines (HAs) were pointed out as components with high carcinogenic potential (Bouvard et al., 2015). from https://doi.org/10.1016/j.meatsci.2016.11.009
- Several works have pointed that HAs can be inhibited by natural antioxidants, such as spices and extracts rich in polyphenols (10.1007/s13197-015-2137-0; https://doi.org/10.1016/j.meatsci.2016.11.009; https://doi.org/10.1021/jf302227b) however the mechanisms undelaying the effect have been overlooked. In the present, work the mechanism of action were explored more in-depth…
- The authors should have given much more information about the cooking patties, such as cooking temperature, time of cooking, cooking losses and about PhIP extraction and quantification, method validation etc. For example, what is the recovery in meat samples and model systems, LOD, LOQ, RSD, R2 values?
- In Conclusions section include the significant level and/or the correlation (line 428; 434).
Author Response
Point 1: In Introduction add the following idea: The International Agency for Research on Cancer (IARC) classified processed meat as “carcinogenic” to humans (group 1), based on > 800 epidemiological studies that reported a link between meat consumption and cancer. Concerning grilled and barbecued meat, heterocyclic aromatic amines (HAs) were pointed out as components with high carcinogenic potential (Bouvard et al., 2015). from https://doi.org/10.1016/j.meatsci.2016.11.009
Response 1: We have rewritten this part according to the Reviewer’s suggestions. Thank you very much for your comments.
Point 2: Several works have pointed that HAs can be inhibited by natural antioxidants, such as spices and extracts rich in polyphenols (10.1007/s13197-015-2137-0; https://doi.org/10.1016/j.meatsci.2016.11.009; https://doi.org/10.1021/jf302227b) however the mechanisms underlying the effect have been overlooked. In the present work, the mechanism of action were explored more in-depth…
Response 2: We have rewritten this part followed the Reviewer’s suggestions. Thank you very much for your comments.
Point 3: The authors should have given much more information about the cooking patties, such as cooking temperature, time of cooking, cooking losses and about PhIP extraction and quantification, method validation etc. For example, what is the recovery in meat samples and model systems, LOD, LOQ, RSD, R2 values?
Response 3: We have added these information to the manuscript according to the Reviewer’s comments. Thank you very much for your comments.
Point 4: In Conclusions section include the significant level and/or the correlation (line 428; 434).
Response 4: We have added the significant level in Conclusions section. Thank you very much for your comments.
Reviewer 4 Report
It is opinion of the reviewer that this interesting and properly prepared paper needs only minor revision. My individual comments are listed below.
Title – A full name of PhIP should be completed. The authors used compounds from Sigma not compounds isolated from TRBE. Therefore, in the title it should be a sentence “main phenolic compounds of…”
21 and entire paper – I suggest to use abbreviation of “TRBE”.
53 – It should be – It should be “added to” instead of “present in”.
59 – “-O-“ must be in italic.85 – It should be just “1307”.
100 – “suspended” instead of “dissolved”.
101 – Amounts/volumes of the sample and reagents should be reported.
102 – How long was extracted.
174 – Correlation should be mentioned in the Statistical analysis section.
Author Response
Point 1: Title – A full name of PhIP should be completed. The authors used compounds from Sigma not compounds isolated from TRBE. Therefore, in the title it should be a sentence “main phenolic compounds of…”
Response 1: The full name of PhIP in Title have been completed. According to the manuscript, the main phenolic compounds in TRE were isorhamnetin, hispidulin, and cirsimaritin, while isorhamnetin and hispidulin inhibited PhIP by trapping phenylacetaldehyde as a key mechanism, the inhibitory mechanism of cirsimaritin on PhIP formation was still unknown. Therefore, we considered it was more proper to use “Isorhamnetin and hispidulin from Tamarix ramosissima” rather than “main phenolic compounds of…”. Thank you very much for your comments.
Point 2: 21 and entire paper – I suggest to use abbreviation of “TRBE”. 53 – It should be – It should be “added to” instead of “present in”. 59 – “-O-“ must be in italic. 85 – It should be just “1307”. 100 – “suspended” instead of “dissolved”. 101 – Amounts/volumes of the sample and reagents should be reported. 102 – How long was extracted.
Response 2: We have made correction according to the Reviewer’s comments. Thank you very much for your comments.
Point 3: 174 – Correlation should be mentioned in the Statistical analysis section.
Response 3: We are very sorry for our negligence of adding the analyzed methods of correlation. Thank you very much for your comments.
Round 2
Reviewer 2 Report
The manuscript was greatly improved.